

# Real-time food allergen detection using OCR-enhanced machine learning techniques

Erol Kına

Özalp Vocational School, Department of Computer Programming, Van Yüzüncü Yıl University, Van, Türkiye

## ABSTRACT

Food allergies are a significant public health concern, emphasizing the need for precise and comprehensive allergen identification in food products. Despite the critical importance of allergen detection, existing allergen food datasets and detection approaches exhibit several limitations. These include small dataset sizes and low accuracy, particularly in real-time scenarios. To address these challenges, this study proposes a novel machine learning-based system evaluated in both real-time and offline environments. The proposed system is designed to analyze ingredient lists extracted from scanned product labels. By leveraging Optical Character Recognition (OCR) technology, the system efficiently retrieves ingredient information in real-time, enabling accurate identification of allergenic components. Once the ingredient information is extracted using OCR, feature extraction techniques such as Bag of Words (BoW), Term Frequency-Inverse Document Frequency (TF-IDF), and Global Vectors for Word Representation (GloVe) are applied. These features play a critical role in training various machine learning and deep learning models. Among the tested models, Logistic Regression (LR) outperformed others, achieving an impressive accuracy of 0.99 with a low computational cost of 13 milliseconds in offline testing. In real-time testing, where product images are captured and processed through the pipeline, the system demonstrated robust performance with a 0.90 accuracy score.

## INTRODUCTION

Food allergies are a group of unfavorable food reactions that might harm a person's health (*White, 1978*). Food allergies are caused by chemicals found in food that are safe to eat but harm the immune system and the human body (*Muthukumar et al., 2020*). A person's immune system essentially triggers these reactions by reacting to particular foods. Serious effects of allergenic content on humans have led to a dramatic increase in concern in recent years. The most crucial thing to understand about allergies and their impacts is their effect on people's health (*Alghafari et al., 2021*). Human responses to food allergies can range widely, from mild symptoms to serious health problems (*Gargano et al., 2021*). According to a Michigan State University survey, one in twenty-five, or about 15 million individuals,

Corresponding author
Erol Kına, erolkina@yyu.edu.tr

has been identified as having food allergies (*Halken et al., 2021*). The study by *Macdougall, Cant & Colver (2002)* describes that 3% of adults and 5% of children in Ireland suffer from food allergies. Allergy symptoms appear around 72 h after eating (*Sicherer & Sampson, 2010*). Food allergies can cause various symptoms, including itching, vomiting, and asthma. Food allergy is among the rare but potentially fatal causes of death. Fewer people in wealthy countries suffer from allergies compared to those in poorer countries. This difference depends on several factors, including bacteria, pollution, clean food, and having the right diet and dietary knowledge (*Lichtenberger, 2023*). It is believed that 1% of people worldwide are at risk of dying from allergies (*Baldo & Pham, 2021*). Thankfully, the fatality rate is quite rare in comparison to other causes, however, it can occasionally be fatal (*Foong et al., 2023*). An allergic response can cause several symptoms, such as swollen lips, itchy skin, diarrhea, and discomfort in the forehead and eyes. One of the most severe allergic reactions that can be fatal is anaphylaxis. Avoidance of allergenic foods is currently the only effective way to prevent allergic reactions.

Allergies to certain foods are incurable. Reading the ingredient labels is always required. The European Union (EU) has distinct regulations governing the accurate and comprehensive labeling of ingredients, including information on allergens (*Herman et al., 2021*). The law was established to ensure that food ingredients are understood and that people are mindful of what they eat. Several rules about allergies specific to a country's cuisine have been put in place. All packaged goods, online products, distance-selling businesses, loose food products, and restaurants are subject to the EU Labelling Directive, which mandates that allergens be labeled using a specific typeface and color. This rule establishes 14 allergens that, if included as ingredients, must be labeled (*Blom et al., 2021*). Celery, wheat-based cereals like barley and oats, and sea creatures such as prawns, crabs, and lobsters are among a list of fourteen allergens. They also include eggs and fish with fins, like tuna or salmon. There is also a lupin-based food product that may cause an allergy. Dairy products like milk are also listed. In some developing countries, people do not have adequate knowledge about food allergies. Even governments have not made sufficient efforts to raise the standard of food safety. Finding allergens in the ingredient list might be quite difficult in these nations.

With its ability to provide novel insights and individualized therapies, machine learning (ML) has become a promising tool for monitoring and forecasting food allergies. Numerous studies have been conducted to address food allergies for example, the study by *Zhang et al. (2023a)* the use of ML in food allergy prediction modeling, which makes use of dietary patterns, genetic markers, and clinical data to accurately forecast the risk of developing certain allergies. Similarly, *Alag (2019)* merged genomic data with machine learning techniques to identify important genetic markers linked to certain food allergies, creating opportunities for precision treatment in this field. Addressing the intricate task of cross-reactivity, the study (*Shanthappa & Kumar, 2022*) used machine learning-powered predictive analytics to identify and forecast cross-reactivity patterns, enhancing allergy prevention and diagnostic precision. Furthermore, *Johnson et al. (2011)* used machine learning algorithms to combine food habits, environmental variables, and clinical history to provide a personalized risk assessment.

## Limitations and gaps

The study on the food allergen dataset faces several limitations in the literature. Many datasets provide only basic information, such as food product names, primary ingredients, and allergens, while often lacking comprehensive data on sweeteners, fats, oils, and seasonings, which are crucial for understanding allergenic potential. Additionally, the datasets may oversimplify predictions by labeling food products without accounting for complex interactions between ingredients and allergens, limiting their applicability in diverse real-world scenarios. This study presents a novel machine learning-based system designed to read ingredient lists from scanned product labels. The system analyzes the extracted information to detect the presence of allergenic components, thereby supporting the prevention of allergic reactions and promoting public health. The proposed system takes food images as input, detects the presence or absence of allergens, and provides detailed information about identified allergens. It is designed for seamless integration with existing platforms, such as mobile applications. The study evaluates the performance of various machine learning classifiers using different feature extraction methods. The main contribution points of this study are:

- Development of an innovative machine learning-based system capable of analyzing ingredient lists from scanned product labels to detect the presence of allergenic components, contributing to the prevention of allergic reactions and promoting public health.
- Integration of Optical Character Recognition (OCR) for real-time extraction of ingredient information from product labels, ensuring accurate and efficient data retrieval.
- Proposal of a novel machine learning approach leveraging Logistic Regression (LR) for rapid and accurate detection of allergens, demonstrating superior performance in predicting allergic and non-allergic elements.
- Design of a system for seamless integration with existing platforms, such as mobile applications, enabling user-friendly access to allergen information for enhanced consumer safety.

The remainder of the article is organized as follows: 'Related Work' reviews related work and discusses key findings from previous research. 'Proposed Methodology' introduces the proposed system architecture, outlines the requirements, and summarizes the machine learning approaches used for food allergy detection. 'Proposed Methodology' details the dataset and methodological strategies for data preprocessing and feature engineering. The experiments and their outcomes are documented in 'Results and Discussion', while 'Conclusions' presents the conclusions drawn from the research and offers recommendations for future work.

## RELATED WORK

The literature on food allergy classification and detection is discussed in this section, which focuses on allergy identification; however, comparable issues with natural language

**Table 1 Summary of literature review.**

| Reference | Year | Classifier(s) | Dataset(s) | Type | Accuracy | Limitation(s) |
|---|---|---|---|---|---|---|
| Ktona et al. (2022) | 2022 | Part, C4.5 | Medical laboratory intermedia | ML | 85.99% | Small number of examples and a limited number of allergy features. |
| Kumar & Rana (2023) | 2023 | Ensemble (ETC + DBN + CatBoost), LR, KNN, PSVM, RF, ADA, GBC, MLP, DNB, ETC, XGB, SVM with RBF | IUIS, SDAP, PDB | ML, DL | 89.16% | Limited dataset diversity and small dataset size. |
| Omurca et al. (2019) | 2019 | DT, KNN, SVM, LR, RF, ETC, Ensemble (DT + SVM + LR + KNN) | Kocaeli University Research and Application Hospital | ML | 77% | Does not address class imbalance. |
| Nedyalkova et al. (2023) | 2023 | KNN, RF (embedded), SVM, NB | CSL, FARRP, SDAP | ML | 93% | Structural and physicochemical parameters affect results. |
| Wang et al. (2021) | 2021 | LGBM, XGB, RF, BERT | Allergen Nomenclature, SDAP, NCBI | ML, DL | 93.10% | Time-consuming (19,500 s); limited allergen data. |
| Sharma et al. (2021) | 2021 | RF, SVC, KNN, XGB, LR, GNB, DT | IEDB | ML, DL | 83.39% | Applicable to chemical reactions, not food. |
| Roither, Kurz & Sonnleitner (2022) | 2022 | KNN, LR, SVC, DT, Linear SVC, MLP | Open food facts dataset | ML, DL | 93% | Does not assess allergenicity ratings. |
| Mishra et al. (2022) | 2022 | YOLOv2, YOLOv3, YOLOv4, YOLOv5, YOLO-R | Allergen30 | ML | 78.29% | Image-only dataset. |
| Rohini et al. (2021) | 2021 | CNN, InceptionV3, VGG16, ResNet50, Xception, VGG19, Inception, ResNetV2, MobileNetV2 | Kaggle Fruits 360 | DL | 97.37% | Limited to fruits and packaged foods; not generalizable to vegetables or dry fruits. |
| Yang et al. (2020) | 2020 | RF with doc2vec | VirusMINT | ML | 87% | No integration of structural or network features. |
| Metwally et al. (2019) | 2019 | LSTM, SVM, MLP, ANN | DIABIMMUN | ML, DL | 67% | Small training set, few time points, poor clinical prediction. |
| Zhang et al. (2023b) | 2023 | RoBERT | BIOPEP-UWM | DL | 97.85% | Relies on in silico predictions and molecular docking. |
| Akbar et al. (2023) | 2023 | XGB, ETC, SVM, ADA, FKNN, LGBM | PredNeuroP | ML | 94.47% | Relies on GA-based ensemble learning. |

processing arise when classifying foods, and Table 1 summarizes key studies addressing these challenges through ensemble machine learning techniques for allergen detection and peptide-based analysis.

Consequently, a thorough review of allergen food classification articles aids in resolving issues with allergen identification. *Ktona et al. (2022)* conducted a study aimed at assisting allergic patients in Albania and proposed a machine learning system for diagnosing the presence of allergic items. Two machine learning models, PART and C4.5, were used. The dataset used in this study was relatively small, consisting of only 155 instances. The results showed that the PART achieved the highest accuracy of 85.99%. Similarly, *Kumar & Rana (2023)* proposed a deep learning-based system for the classification of protein allergens. In this study, a group of models based on deep learning was used to find identify allergens using CatBoost, ETC+DBN. In 2020, the new group model worked better by joining the

predictions of three models using voting criteria. They also tested how well their learning models were working by using them individually. The study's results show that the proposed ensemble model did better than other learning models and achieved an accuracy of 89.16%. Furthermore, *Omurca et al. (2019)* developed a smart assistant to predict the type of allergy in Turkey without human assistance. They used standard machine learning methods such as decision trees (DT), LR, support vector machines (SVM), and k-nearest neighbors (KNN), along with ensemble classifiers. This study is limited to Turkey and got allergy data from Kocaeli University Research & Application Hospital. The findings of the study showed the highest accuracy score of 77% using the majority voting method for recognizing 18 distinct allergies.

Similarly, in further work on protein-based food allergies, the study by *Nedyalkova et al. (2023)* developed a machine learning-based system capable of detecting allergies caused by proteins. This was achieved using various labels and ML methods. The study utilized a dataset containing both allergenic and non-allergenic proteins. This was done using different labels and ML methods. The study used a set of data that included both allergen and normal proteins. In this study, they selected the important features to describe them properly. To check the performance of the learning models, they used different evaluation parameters and validated the performance of the ML algorithms. The results of the study showed that KNN outperformed the other learning models in the classification of allergen foods. However, SVM also performed better when using fewer descriptors. The study showed that using ML methods could help predict protein-related allergy risk. It also emphasized that choosing the right features could help attain better results. Another study by *Wang et al. (2021)* presented a deep learning model with self-attention for predicting allergenic foods. Additionally, they applied traditional machine learning models such as Light Gradient-Boosting Machine (LGBM), eXtreme Gradient Boosting (XGB), and Random Forest (RF) to address the same problem. To underscore the effectiveness of their proposed approach, various commonly used machine learning models were chosen as baseline classifiers in this study. The findings revealed that Bidirectional Encoder Representations from Transformers (BERT) achieved the highest accuracy score of 93.10%. Furthermore, a comparison with their previous study demonstrated that their proposed model surpassed the results of the earlier investigation.

Furthermore, the study by *Sharma et al. (2021)* used molecular data and machine learning methods to develop a system that can predict how chemical compounds might cause allergies. The PaDEL program was used to calculate molecular details for 403 allergenic and 1,074 non-allergenic substances in the Immune Epitope Database (IEDB) database. The models were trained on a dataset that included 2D and 3D samples, and then Fingerprint (FD) features were tested. The best results were demonstrated by hybrid descriptor RF model. It had an Area Under the Curve (AUC) of 0.93 and the highest accuracy of 83.39% on the test dataset. The study found that some chemical markers, like GraphFP1014 and PubChemFP129, were common in allergy-causing substances. The study by *Roither, Kurz & Sonnleitner (2022)* proposed an automated system for the classification of the allergen and style in different recipes. The data used had labels for the top 14 allergen groups. Results of the study showed that LR with

Term Frequency-Inverse Document Frequency (TF-IDF) features achieved an accuracy score of 93% for the milk-containing items. Similarly, the study by *Mishra et al. (2022)* proposed a deep learning-based computer vision system to detect food items with possible food allergens. In this study, they used different variants of YOLO on the Allergen30 dataset. The result of the study shows that the YOLO-R achieved the highest F1-score among all variants which is 78.29% for image classification. Further, the study by *Rohini et al. (2021)* proposed a system for automatically sorting allergens and nutrients in fruits. They also looked at packaged food using the same method. They used many types of deep learning models like convolutional neural network (CNN), InceptionV3, Visual Geometry Group (VGG)16, and Visual Geometry Group (VGG)19. They included built-in models called Inception ResNetV2, ResNet50, Xception, and MobileNetV2 in this study. The result of the study shows that the VGG16 outperformed the other learning models and achieved an accuracy score of 97.37%.

The study by *Yang et al. (2020)* worked on human-virus protein-protein interactions using an RF classifier powered by doc2vec embeddings. According to the findings, this computational framework outperformed various combinations of well-known machine learning algorithms and sequence encoding techniques. By using feature embedding for protein representation, the method captured more contextual information from protein sequences, considerably improving prediction performance. The authors argued that their findings could help identify potential interactions between human and viral proteins as well as guide experimental attempts to discover proteins implicated in human virus interactions and their associated functions. They also suggested that future advancements in deep learning architectures, protein structural data, and host Protein-Protein Interaction (PPI) network topology could significantly improve the accuracy of human virus PPI predictions. The highest accuracy achieved was 87% by RF. The authors (*Metwally et al., 2019*) worked on diagnosing food allergies in early life (0–3 years) and exploring the predictive capacity of Long Short-Term Memory (LSTM) networks for identifying food allergies using longitudinal gut microbiome profiles. The study utilized the DIABIMMUNE dataset, and the LSTM model demonstrated superior performance compared to classical methods such as Hidden Markov Models, Multilayer Perceptron (MLP), Support Vector Machine (SVM), RF, and Least Absolute Shrinkage and Selection Operator (LASSO) regression. To enhance model efficacy, various feature selection and extraction methods were tested, including a sparse autoencoder for latent representation extraction. Another study by *Zhang et al. (2023b)* proposed a novel approach using a protein-specific deep learning model, Protein Bidirectional Encoder Representations from Transformers (ProtBERT), to efficiently screen for antihypertensive peptides. ProtBERT demonstrated superior performance with an AUC of 0.9785 compared to other deep learning models. The model was applied to screen peptides from soybean protein isolate (SPI). Subsequent molecular docking and *in vitro* validation identified three peptides—LVPFGW (IC50 = 20.63 $\mu M$), VSFPVL (2.57 $\mu M$), and VLPF (5.78 $\mu M$)—with strong antihypertensive activity.

Similarly, the authors in the study by *MacMath, Chen & Khoury (2023)* explored the field of AI in allergy and immunology that has the potential to revolutionize diagnostic

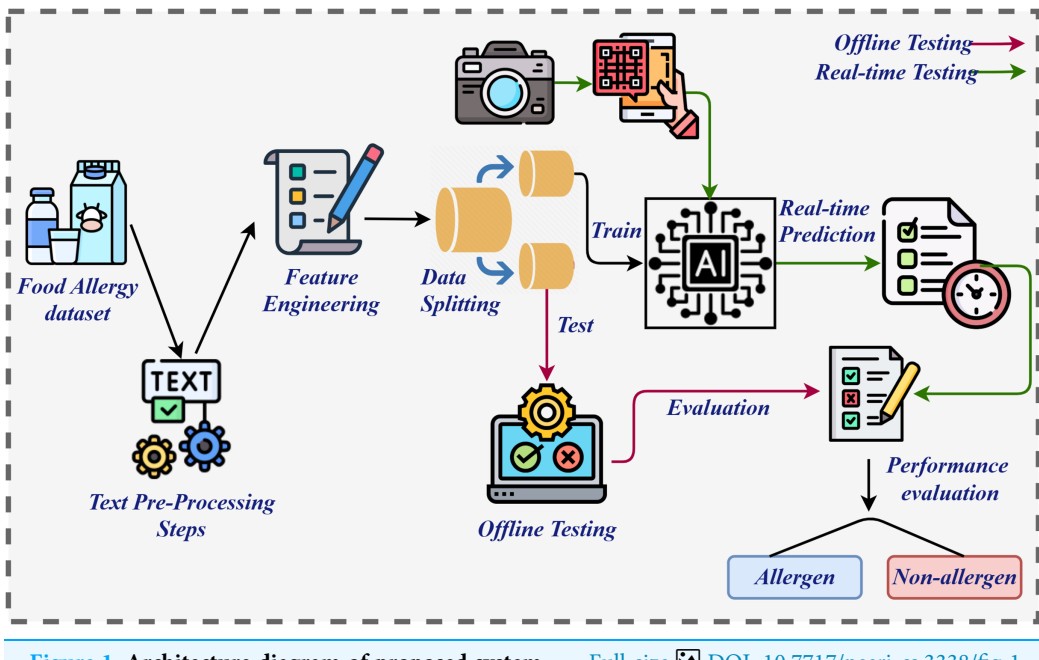

**Figure 1 Architecture diagram of proposed system.**

accuracy, therapeutic strategies, and clinical care. The study described that most ML models in this field have primarily addressed inborn errors of immunity, asthma, and atopic dermatitis; other domains are emerging as areas of exploration. Lastly, authors have (*Akbar et al., 2017*) introduced an ensemble learning approach leveraging genetic algorithms for detecting anticancer peptides, providing a detailed mathematical framework for the method. Their subsequent work expanded this approach to other peptide classes, including antitubercular peptides similarly in another extension of their work (*Akbar et al., 2021*) and neuropeptides (*Akbar et al., 2023*) studies combined multiple machine learning techniques, such as SVM, RF, and KNN, within ensemble learning frameworks, demonstrating their versatility and effectiveness in peptide identification tasks. This body of work highlights the potential of ensemble methods to improve predictive accuracy in bioinformatics and peptide-based therapeutic discovery.

# PROPOSED METHODOLOGY

The primary contribution of this study lies in addressing a significant public health challenge through the development of an advanced, real-time allergen detection system, leveraging machine learning and deep learning technologies. The methodology for the proposed system, illustrated in Fig. 1, began with the collection and pre-processing of a dataset. Pre-processing was a critical step in ensuring data consistency and extracting valuable information, as highlighted by *Shafique et al. (2023)*. This process involved removing punctuation, eliminating stopwords, converting text to lowercase, and applying stemming techniques, resulting in a refined dataset optimized for algorithm training. Subsequently, feature engineering was performed using techniques such as BoW, GloVe, and TF-IDF, each chosen for their ability to extract robust features essential for effective

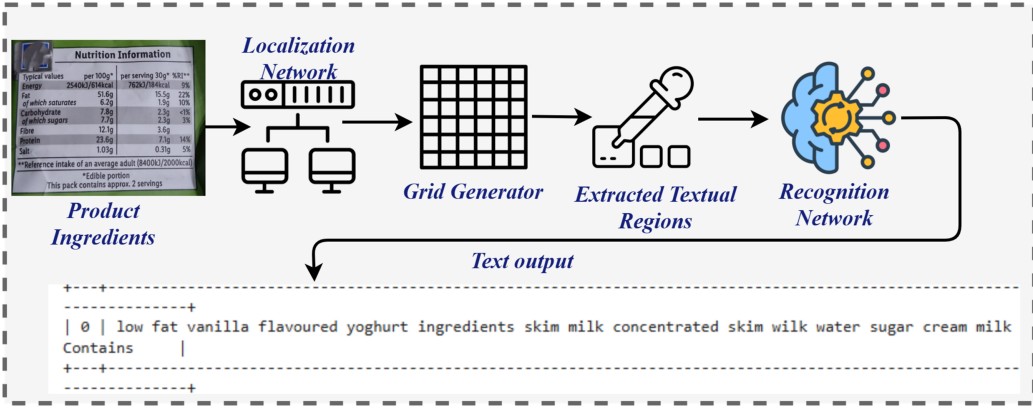

**Figure 2 Architecture diagram for OCR extracting text from product scans.**

training (*Shafique et al., 2024b*). The processed dataset was then divided into training and testing subsets in an 80:20 ratio. Both machine learning and deep learning models were employed for training and evaluation in offline and online testing environments. During offline testing, LR excelled when applied to BoW features, whereas RF performed notably well with TF-IDF features. LR, as a linear model, is particularly effective with high-dimensional and sparse features, such as those in BoW, where the presence or absence of terms provides substantial discriminatory power for binary classification (*Berrar, 2018*). BoW captures straightforward, effective relationships between terms and labels, aligning well with LR's capacity to establish linear decision boundaries. In contrast, RF, as a non-linear ensemble model, thrives on richer, more informative features like TF-IDF (*Bania, 2020*). The weighting of terms by their relative importance in the document *corpus* enables RF to exploit subtle patterns in word frequencies and interactions, enhancing its ability to make more precise decisions.

In the online testing phase, the methodology incorporated a novel approach to allergen detection by analyzing images of food ingredients. Optical Character Recognition (OCR) was employed to extract textual information from the images, facilitating accurate allergen identification as shown in Fig. 2. The diagram illustrates the process for extracting text from product ingredient labels using an OCR system. It begins with an image of the product label, which is processed by a Localization Network to identify regions that are likely to contain text. These regions are further divided into smaller sections using a Grid Generator to create a structure for easier analysis. The textual regions are then highlighted with Bounding Boxes, isolating the areas of interest. The Recognition Network processes these extracted regions, converting the visual text into machine-readable output. Finally, the system outputs the extracted text, providing a structured and readable format, such as a list of product ingredients, which is further processed by applying pre-processing steps and subsequently by extracting features, and then passed to the model for final prediction. This integrated workflow of image capture, analysis, and OCR application significantly enhanced both the precision and efficiency of the detection system. By enabling the

analysis of complex textual details within food ingredient images, OCR further strengthened the system's capabilities, offering a reliable and effective solution for allergen detection. Ultimately, this approach advances the development of allergen detection systems while contributing to public health by mitigating risks associated with allergenic food consumption.

The architecture of the proposed model leverages an LR classifier with a BoW representation. The BoW approach is used to transform the textual data into a format suitable for machine learning algorithms by representing each document as a vector of word frequencies. This enables the LR model to capture and classify text-based features effectively. The chosen method was motivated by the simplicity and effectiveness of LR for binary classification tasks in text analysis. Regarding the design and packages used, the model relies on popular Python libraries such as scikit-learn for logistic regression, sklearn. feature_extraction.text.CountVectorizer for BoW, and other supporting libraries like pandas for data handling.

The CountVectorizer is utilized to convert text into feature vectors and is a fundamental part of the feature engineering process for the model. All packages are well-established in the machine learning community, ensuring robustness and reliability of the results. As for the hyperparameters, for the LR model, parameters such as the regularization strength were set based on standard practices in the field. These parameters, including the solver and max iterations, are chosen to avoid overfitting while ensuring that the model converges efficiently. The fixed values were selected after preliminary testing and are based on the characteristics of the dataset. Though auto-tuning could be performed for further optimization, the focus on fixed parameters here was intended to provide a baseline model for comparison. If necessary, we could implement hyperparameter tuning techniques such as grid search or random search to optimize these parameters dynamically. This approach was chosen because logistic regression with BoW is a well-tested, interpretable solution, and setting the parameters initially allowed us to evaluate the performance before applying complex tuning methods.

## Dataset

The dataset of food ingredients and allergies is sourced from *Kaggle Dat (2023)*. The dataset consists of a collection of information regarding allergens present in various food items. The data contain information about allergy-related ingredients in different food items. The dataset consists of 400 records and seven attributes, each representing a specific food item and its associated allergens. A detailed list of all the allergens present in the food products is included in the dataset. These allergens include a broad spectrum of foods, including dairy, wheat, and nuts (almonds, peanuts, pine nuts), seafood (anchovies, fish, shellfish), grains (oats, rice), ingredients derived from plants (celery, mustard, soybeans), animals (pork, poultry), and common components (cocoa, eggs). Additionally, the dataset contains entries where no specific allergens are listed. Dataset attributes and their description are given in Table 2 and sample size and counts are shown in Table 3.

**Table 2  Key features of the allergen food dataset.**

| Attribute | Description |
|---|---|
| Food product | Contains information about food products. |
| Main ingredient | Describes the primary food ingredients. |
| Sweetener | Indicates the type or presence of a sweetener used. |
| Fat/Oil | Specifies the type or presence of fat or oil used. |
| Seasoning | Lists spices or seasonings added to enhance flavor. |
| Allergens | Lists allergens associated with the product, indicating potential allergic reactions. |
| Prediction | Label for the food product based on ingredients and allergens. |

**Table 3  Class distribution and sample sizes.**

| Class | Total *corpus* | Training samples | Testing samples online | Testing samples offline |
|---|---|---|---|---|
| Contains | 400 | 320 | 20 | 80 |
| Does not contain | 400 | 320 | 20 | 80 |

**Table 4  Comparison of raw and preprocessed data samples.**

| Before preprocessing | After preprocessing |
|---|---|
| Salad cucumbers olive oil tomato, onion | Salad cucumb oliveoil tomato onion |
| Flavored yogurt yogurt sugar raspbery dairy | Flavor yogurt yogurt sugar raspberi dairy |
| Peanuts salad peanuts olive oil spices | Peanut salad peanut oliveoil spice |

**Table 5  Sample data aggregation.**

| | Food product | Main ingredient | Sweetener | Fat/Oil | Seasoning | Allergens |
|---|---|---|---|---|---|---|
| Original data | Almond cookies | Almonds | Sugar | Butter | Flour | Almonds, Wheat, Dairy |
| After data aggregation | Almond cookies, Almonds sugar, Butter flour, Almonds, Wheat, Dairy | | | | | |

**Table 6  Removing punctuation marks from the text.**

| Before preprocessing | After preprocessing |
|---|---|
| Almond cookies almonds | Almond cookies almonds |
| Sugar butter flour almonds, Wheat, Dairy | Sugar butter flour almonds wheat dairy |

## Data preprocessing

Data preparation is a crucial step that aims to increase the accuracy of the model, facilitate the processing of data by model, and enhance data quality (*Shafique et al., 2024a*). The food allergy dataset includes unstructured text data, containing elements such as stemming, stopwords, punctuation, and capital and lowercase characters. Text pre-processing is a vital step in the process of standardizing raw data. Table 4 shows some samples before and after applying the preprocessing steps. As shown in Table 5, individual food attributes such as sweeteners and allergens are merged into a unified representation.

**Table 7  Removing stopword removal from data.**

| Before preprocessing | After preprocessing |
| --- | --- |
| Almond cookies and butter with sugar and flour | Almond cookies butter sugar flour |
| This is a yogurt with dairy and raspberry flavor | Yogurt dairy raspberry flavor |

**Table 8  Transforming text case for dataset uniformity.**

| Before preprocessing | After preprocessing |
| --- | --- |
| Almond cookies almonds | Almond cookies almonds |
| Sugar butter flour almonds wheat dairy | Sugar butter flour almonds wheat dairy |

**Table 9  Stemming transformation for standardizing dataset terms.**

| Before preprocessing | After preprocessing |
| --- | --- |
| Almond cookies almonds | Almond cooki almond |
| Sugar butter flour almonds wheat dairy | Sugar butter flour almond wheat dairy |

### Punctuation removal

Punctuation marks and symbols like @,!,$, %, " are eliminated from the text. Punctuation removal is the process of removing all commas, symbols, and exclamation points. Text before and after the punctuation removal is shown in Table 6.

### Stop-word removal

Stop words are eliminated from the dataset in the next stage. Stop words improve the readability of sentences but they are not useful for text analysis. Eliminating stop words improves the classification algorithm's efficiency (*Shafique, Mehmood & Choi, 2019*). In stop word removal, terms are discarded that do not contribute to the sense of the statement, such as "and," "the," "is," and "a." The text's grammatical and syntactical structure does not improve the model's accuracy. The model's accuracy is reduced when stop words appear often in a sentence. The dataset's dimensions are reduced, which is the primary goal for training machine learning models more efficiently. Secondly, it is important to use only those terms that influence decision-making and content. Text before and after the stopword removal is shown in Table 7, and to achieve uniformity, all input text is converted to lowercase, as demonstrated in Table 8.

### Stemming

The technique of stemming involves removing word affixes and reducing words to their root forms. For instance, salty, salts, and salted are synonyms for salt that have the same meaning. Removing the suffixes enhances classifier learning capacity and lowers feature complexity. To illustrate that a sample from the dataset is shown in Table 9.

## Feature engineering methods

Feature engineering is the process of selecting, creating, or transforming features (input variables) in a dataset to improve the performance of machine learning models

(*Raza et al., 2022*). It involves identifying relevant information from raw data and representing it in a format that is suitable for predictive modeling. This is done to help programs learn in a machine-learning system. Algorithms for machine learning can perform better when feature engineering is used. "Trash in, trash out" is a machine-learning proverb that is frequently used. According to this, the meaningless input leads to a meaningless output. However, more informative data could also lead to desired outcomes. Consequently, feature engineering may be used to extract relevant features from unprocessed data, hence improving the accuracy and consistency of learning algorithms. We employed three feature engineering techniques in this study: BoW, GloVe, and TF-IDF.

### BoW

BoW is an easy-to-understand technique for extracting features from text data (*Radovanović & Ivanović, 2008*). BoW is highly useful in problems such as text classification and language modeling. This approach extracts features by using a Count Vectorizer. The CountVectorizer creates a sparse token matrix by counting the occurrences of tokens, or term frequencies. BoW is a set of words and features, with a value assigned to each feature to indicate how often that feature occurs.

### GloVe

GloVe, which stands for Global Vectors for Word Representation, is a widely used technique in natural language processing (NLP) for generating word embeddings. These embeddings depict semantic connections among words in a given text *corpus* by representing each word as a compact vector within a continuous vector space. The GloVe algorithm trains word embeddings by analyzing the overall co-occurrence patterns of words in a *corpus* (*Aljedaani et al., 2022*). It constructs a co-occurrence matrix, where each entry reflects the frequency of two words appearing together within a specified context window. By decomposing this co-occurrence matrix, GloVe extracts word embeddings that encode semantic similarities between words. These GloVe embeddings, once learned, serve as valuable features in various NLP tasks such as text categorization, sentiment analysis, and machine translation. They encapsulate semantic associations among words, enabling models to grasp the underlying meaning of texts and perform effectively across diverse tasks and domains.

### TF-IDF

Another method used for feature extraction is TF-IDF. TF-IDF involves transforming words or phrases into numbers that machines can understand and use for various tasks such as searching and analyzing patterns in text documents (*Qaiser & Ali, 2018*). Text analysis and information retrieval are the two main applications of TF-IDF. Based on term frequency (TF) and inverse document frequency (IDF), TF-IDF assigns a weight to every term in the document. The importance of the terms is based on the frequency of this specific term. Using the following formula in Eq. (1) TF-IDF calculates the weight of each term.

**Table 10  Hyperparameters and tuning values range.**

| Model | Parameters tuning |
|---|---|
| LR | Default settings |
| NB | Default settings |
| KNN | n_neighbour = 3 |
| RF | n_estimators = 300, random_state = 2, max_depth = 100 |
| CNN | Conv1D (filter = 32, 64), optimizer = adam, loss = binary_crossentropy, Dropout = 0.5, activation = 'relu', epochs = 100 |
| GRU | Conv1D (filter = 16, 32, 64), optimizer = adam, loss = binary_crossentropy, Dropout = 0.5, activation = 'relu', epochs = 100 |
| LSTM | Conv1D (filter = 16, 32, 64), optimizer = adam, maxpooling1D = 1 × 1, loss = binary_crossentropy, Dropout = 0.5, activation = 'relu', epochs = 100 |

$$W_{i,j} = \frac{TF_{i,j}}{N/D_{f,t}}. \tag{1}$$

Here, $N$ represents the total number of documents in a *corpus*. $TF_{i,j}$ denotes how many times the term $t$ appears in document $d$, and $Df_t$ represents the number of documents that contain term $t$.

## Machine learning model

In this section, we discuss the machine learning models used for allergen food detection. Machine learning techniques were implemented using Natural Language Toolkit (NLTK) and the Scikit-learn modules. The Sci-Kit module was utilized to implement the four supervised machine-learning algorithms using Python. Regression and classification issues are frequently resolved using supervised machine-learning techniques. A linear model is called LR, whereas a tree-based technique, RF is used. Other methods used to check the efficacy of the proposed system are KNN and Naive Bayes (NB) (probability). Table 10 displays the implementation details of the algorithms along with their hyperparameters.

The hyperparameter settings for the algorithms highlight a balance between simplicity and optimization tailored to the specific models. LR and NB use default settings, reflecting their robustness without extensive tuning. KNN employs n_neighbors = 3 to capture fine details in the data. For RF, the parameters include n_estimators = 300, random_state = 2, and max_depth = 100 to enhance model stability and prevent overfitting. Deep learning models such as CNN, GRU, and LSTM share advanced settings, including multiple filter sizes (16, 32, 64), the Adam optimizer for efficient learning, binary_crossentropy as the loss function, Dropout = 0.5 for regularization, and epochs = 100 for sufficient training. LSTM further incorporates maxpooling1D = 1 × 1 for feature selection. These configurations reflect a strategic approach to leveraging each algorithm's strengths for optimal performance.

### OCR

OCR (*Karthick et al., 2019*) plays a crucial role in identifying food allergens. It extracts useful text from images of food labels containing ingredient information. When images are

taken, OCR methods are used to clean up and spot letters allowing for the extraction of information about ingredients and allergy warnings. This extracted text is the key for training machines to identify and classify possible allergens. This process enhances the accuracy and reliability of food allergy detection systems. Connecting OCR with machine learning helps create a complete way to find food allergies. OCR helps change picture text into useful data. This makes it easy to use this information in ways that spot allergens more accurately. The team-up of OCR and machine learning helps make good systems for keeping people safe from food allergies.

## Evaluation method

The proposed model was evaluated using a two-fold approach. First, its performance was benchmarked against several state-of-the-art machine learning algorithms, including RF, KNN, LR, and NB, utilizing feature representations such as TF-IDF, BoW, and GloVe. Additionally, we explored deep learning architectures including LSTM, GRU combined with Recurrent Neural Network (RNN), CNN integrated with LSTM, and a hybrid model combining Gated Recurrent Unit (GRU), LSTM, and RNN. Second, the system underwent real-time testing on an independent dataset comprising 200 previously unseen food product images captured using a mobile camera, to assess its effectiveness in detecting allergens under varying real-world conditions.

### *Accuracy*

Accuracy is a significant and frequently used parameter for evaluating how well models perform. It represents the percentage of correctly predicted instances out of the total number of cases. It can be calculated using the following formula.

$$Accuracy = \frac{TP + TN}{TP + TN + FP + FN}. \tag{2}$$

### *Precision*

It is the classifier's exactness. The precision measure is the proportion of positively anticipated cases to all positive instances. It may be computed using the formula that follows:

$$Precision = \frac{TP}{TP + FP}. \tag{3}$$

### *Recall*

The classifier's completeness is measured by recall. It displays the proportion of accurately identified true positive cases. It is computed as:

$$Recall = \frac{TP}{TP + FN}. \tag{4}$$

### *F1-score*

It is seen as a model's well-balanced and well-represented performance as it incorporates both accuracy and recall. The harmonic mean of recall and precision is the F1-score. It is computed using.

**Table 11  Overview of the experimental setup.**

| Element | Details |
|---|---|
| RAM | 16 GB |
| OS | 64-Bit Windows 11 |
| Language | Python 3.8 |
| CPU | Core i7, 7th Generation with 2.8 GHz processor |
| GPU | Nvidia, 1,060, 8 GB |

$$F1\text{-}score = 2 \times \frac{Precision \times Recall}{Precision + Recall}. \tag{5}$$

### Micro average

The micro-average computes the global counts of true positives, false positives, and false negatives across all classes. It treats every prediction equally, regardless of class imbalance.

$$\text{Micro Precision} = \frac{\sum_{c=1}^{C} TP_c}{\sum_{c=1}^{C} (TP_c + FP_c)}. \tag{6}$$

$$\text{Micro Recall} = \frac{\sum_{c=1}^{C} TP_c}{\sum_{c=1}^{C} (TP_c + FN_c)}. \tag{7}$$

$$\text{Micro F1} = \frac{2 \times \text{Micro Precision} \times \text{Micro Recall}}{\text{Micro Precision} + \text{Micro Recall}}. \tag{8}$$

### Weighted average

Weighted-average computes the metric independently for each class and then calculates the average weighted by the number of true instances (support) in each class.

$$\text{Weighted Precision} = \frac{\sum_{c=1}^{C} (\text{Precision}_c \times \text{Support}_c)}{\sum_{c=1}^{C} \text{Support}_c}. \tag{9}$$

$$\text{Weighted Recall} = \frac{\sum_{c=1}^{C} (\text{Recall}_c \times \text{Support}_c)}{\sum_{c=1}^{C} \text{Support}_c}. \tag{10}$$

$$\text{Weighted F1} = \frac{\sum_{c=1}^{C} (\text{F1}_c \times \text{Support}_c)}{\sum_{c=1}^{C} \text{Support}_c}. \tag{11}$$

Here, $TP_c$, $FP_c$, $FN_c$ refer to the true positives, false positives, and false negatives for class $c$, and $Support_c$ denotes the number of actual instances in class $c$. $C$ is the total number of classes.

## RESULTS AND DISCUSSION

The results of several classifiers used to identify allergens and food items are presented in this section. The experiments were conducted on a machine equipped with a 7th-generation Core i7 CPU and running Windows 11 on a Jupyter Notebook, with the ML models constructed using Python 3.8. The learning models' performance was evaluated

**Table 12 Machine learning performance using TF-IDF features.**

| Classifier | Wall time | Accuracy | Class | Precision | Recall | F1-score |
|---|---|---|---|---|---|---|
| RF | 165 ms | 0.99 | Contains | 0.98 | 1.00 | 0.99 |
| | | | Does not Contains | 1.00 | 0.96 | 0.98 |
| | | | Micro Avg. | 0.99 | 0.98 | 0.99 |
| | | | Weighted Avg | 0.99 | 0.99 | 0.99 |
| KNN | 13.5 ms | 0.87 | Contains | 0.94 | 0.86 | 0.90 |
| | | | Does not Contains | 0.73 | 0.88 | 0.80 |
| | | | Micro Avg. | 0.84 | 0.87 | 0.85 |
| | | | Weighted Avg | 0.88 | 0.87 | 0.87 |
| LR | 14 ms | 0.96 | Contains | 0.95 | 1.00 | 0.97 |
| | | | Does not ConTains | 1.00 | 0.88 | 0.94 |
| | | | Micro Avg. | 0.97 | 0.94 | 0.96 |
| | | | Weighted avg | 0.97 | 0.96 | 0.96 |
| NB | 8.02 ms | 0.95 | Contains | 0.95 | 0.98 | 0.97 |
| | | | Does not Contains | 0.96 | 0.88 | 0.92 |
| | | | Micro Avg. | 0.95 | 0.93 | 0.94 |
| | | | Weighted Avg | 0.95 | 0.95 | 0.95 |

through the use of F1-score, accuracy, precision, and recall metrics. Table 11 provides a detailed description of the hardware and software specs used in the experiment.

## Machine learning performance using TF-IDF features

Table 12 highlights the performance of machine learning models using TF-IDF features, with RF achieving the highest accuracy of 0.99, followed by LR at 0.96, NB at 0.95, and KNN at 0.87. TF-IDF enhances feature representation by emphasizing word importance, benefiting RF, LR, and NB with their ability to leverage weighted features effectively, while KNN struggles due to the high-dimensional feature space. RF excels due to its capacity to capture intricate relationships, while LR and NB show strong balanced metrics. Overall, TF-IDF improves model performance significantly, except for KNN, which faces limitations in handling high-dimensional data.

## Machine learning models results using BoW features

Table 13 presents the results of machine learning models using BoW features, with LR achieving the highest accuracy (0.99), followed by RF (0.98), NB (0.96), and KNN (0.88). BoW transforms text into fixed-length vectors based on word frequencies, and this feature extraction technique benefits models like LR and RF, which effectively leverage frequency-based feature representations. RF performs strongly, achieving high precision and recall, especially for the "Contains" class. LR also performs exceptionally, with balanced precision and recall across both classes. KNN experiences performance degradation due to the sparsity and high dimensionality of BoW features, while NB performs decently with relatively balanced metrics. BoW improves performance for linear and tree-based models but presents challenges for distance-based models like KNN.

**Table 13 Machine learning performance using BoW features.**

| Classifier | Wall time | Accuracy | Class | Precision | Recall | F1-score |
|---|---|---|---|---|---|---|
| RF | 125 ms | 0.98 | Contains | 1.00 | 0.96 | 0.98 |
| | | | Does not Contains | 0.93 | 1.00 | 0.96 |
| | | | Micro Avg. | 0.96 | 0.98 | 0.97 |
| | | | Weighted Avg | 0.98 | 0.98 | 0.98 |
| KNN | 11 ms | 0.88 | Contains | 0.96 | 0.86 | 0.91 |
| | | | Does not Contains | 0.74 | 0.92 | 0.82 |
| | | | Micro Avg. | 0.85 | 0.89 | 0.86 |
| | | | Weighted Avg | 0.89 | 0.88 | 0.88 |
| LR | 13 ms | 0.99 | Contains | 1.00 | 0.98 | 0.99 |
| | | | Does not Contains | 0.96 | 1.00 | 0.98 |
| | | | Micro Avg. | 0.98 | 0.99 | 0.99 |
| | | | Weighted Avg | 0.99 | 0.99 | 0.99 |
| NB | 9 ms | 0.96 | Contains | 0.97 | 0.98 | 0.97 |
| | | | Does not Contains | 0.96 | 0.92 | 0.94 |
| | | | Micro Avg. | 0.96 | 0.95 | 0.96 |
| | | | Weighted Avg | 0.96 | 0.96 | 0.96 |

**Table 14 Machine learning performance using GloVe features.**

| Classifier | Wall time | Accuracy | Class | Precision | Recall | F1-score |
|---|---|---|---|---|---|---|
| RF | 192 ms | 0.79 | Contains | 0.84 | 0.86 | 0.85 |
| | | | Does not Contains | 0.67 | 0.64 | 0.65 |
| | | | Micro Avg. | 0.76 | 0.75 | 0.75 |
| | | | Weighted Avg | 0.79 | 0.79 | 0.79 |
| KNN | 14 ms | 0.74 | Contains | 0.83 | 0.70 | 0.81 |
| | | | Does not Contains | 0.57 | 0.64 | 0.60 |
| | | | Micro Avg. | 0.70 | 0.71 | 0.71 |
| | | | Weighted Avg | 0.75 | 0.74 | 0.75 |
| LR | 23 ms | 0.77 | Contains | 0.81 | 0.88 | 0.84 |
| | | | Does not Contains | 0.65 | 0.52 | 0.58 |
| | | | Micro Avg. | 0.73 | 0.70 | 0.71 |
| | | | Weighted Avg | 0.76 | 0.77 | 0.76 |
| NB | 11 ms | 0.67 | Contains | 0.77 | 0.75 | 0.76 |
| | | | Does not Contains | 0.46 | 0.48 | 0.47 |
| | | | Micro Avg. | 0.61 | 0.62 | 0.62 |
| | | | Weighted Avg | 0.67 | 0.67 | 0.67 |

## Machine learning performance using GloVe features

Table 14 presents the results of machine learning models using GloVe features, with RF achieving the highest accuracy of 0.79, followed by LR at 0.77, KNN at 0.74, and NB at 0.67. GloVe provides dense, pre-trained word embeddings that capture semantic relationships between words, which benefits models like RF and LR. However, the

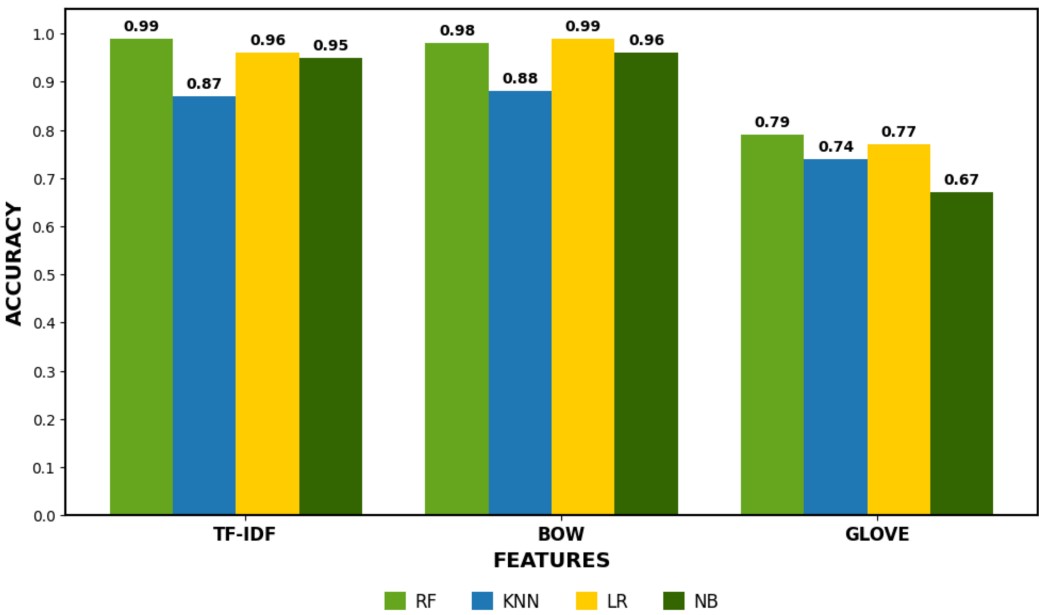

**Figure 3 Machine learning models performance comparison.**

performance with GloVe features is lower than with other techniques such as TF-IDF or BoW. RF performs relatively well but struggles with class imbalance, showing lower recall for the "Does not Contain" class. LR exhibits similar performance, with a good precision-recall trade-off for "Contains" but struggles for "Does not Contain." KNN's performance is also hindered by the high-dimensional and dense nature of GloVe embeddings, while NB performs poorly, especially with the "Does not Contain" class. Although GloVe improves semantic understanding, but its impact on accuracy is less pronounced for these models than simpler methods like TF-IDF or BoW. The results shown in Fig. 3 compare the accuracy of three different feature extraction methods: TF-IDF, BoW, and GloVe. TF-IDF and GloVe achieved the highest accuracy, consistently scoring around 0.95–0.99, indicating their effectiveness in capturing features for classification tasks. In contrast, the BoW method showed lower accuracy, ranging between 0.67 and 0.88, reflecting its limitations in capturing semantic relationships between words. This highlights the strength of TF-IDF and GloVe in delivering better performance for tasks involving natural language understanding.

Figure 4 illustrates a LR training set, where each point represents a sample with two features plotted on the x and y axes. The purple points correspond to the positive class (labeled as "Contains"), while the green points represent the negative class ("Does not contain"). LR uses this labeled data to learn a decision boundary that separates the two classes by estimating the probability that a new point belongs to the positive class. The model aims to find the best line (or curve) that clearly distinguishes purple points from green ones, thereby enabling it to predict whether future, unseen samples are more likely to "contain" or "not contain" the target attribute.

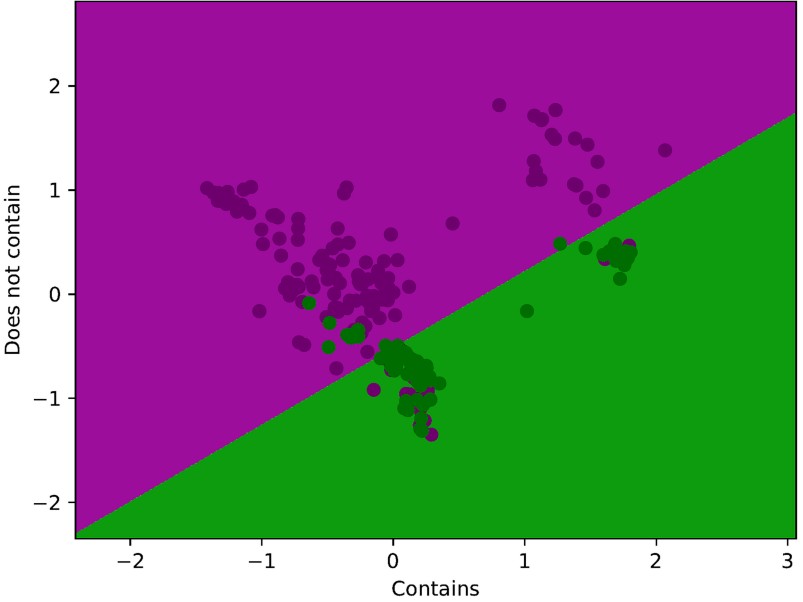

**Figure 4 Visualization of proposed method training data with decision boundary.**

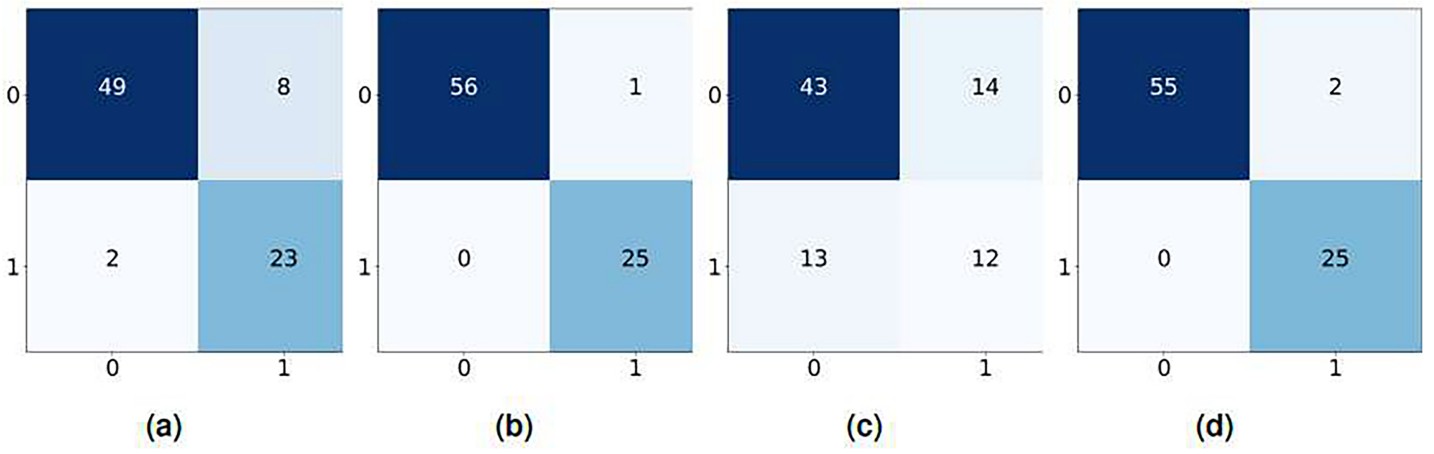

**Figure 5 Confusion matrix using BoW features, (A) KNN, (B) LR, (C) NB, and (D) RF.**

In comparison, the BoW based confusion matrices demonstrate slightly lower performance, though they still reflect reliable classification results. While RF with BoW remains highly accurate, NB exhibits a higher rate of false positives and false negatives, suggesting that BoW may not capture text features as effectively as TF-IDF, as shown in Fig. 5. The confusion matrices for TF-IDF show strong classifier performance, with high true positive rates and minimal false negatives across all models. For example, the RF classifier with TF-IDF achieves 57 true positives and just one false negative, indicating near-perfect classification. LR also performs well, with only a few misclassifications as shown in Fig. 6.

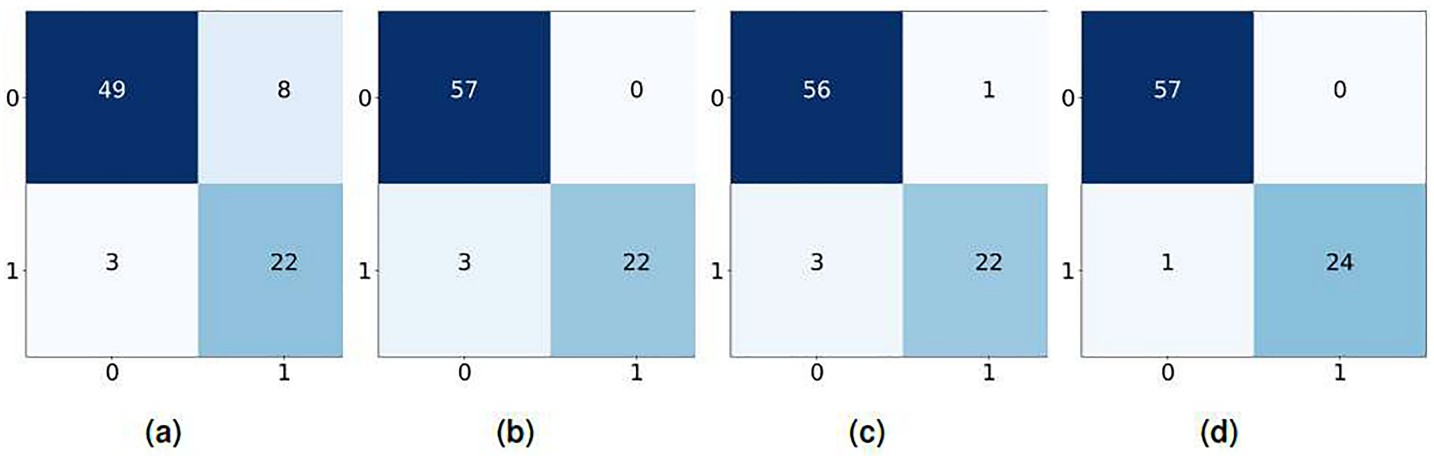

**Figure 6** Confusion matrix using TF-IDF features, (A) KNN, (B) LR, (C) NB, and (D) RF.

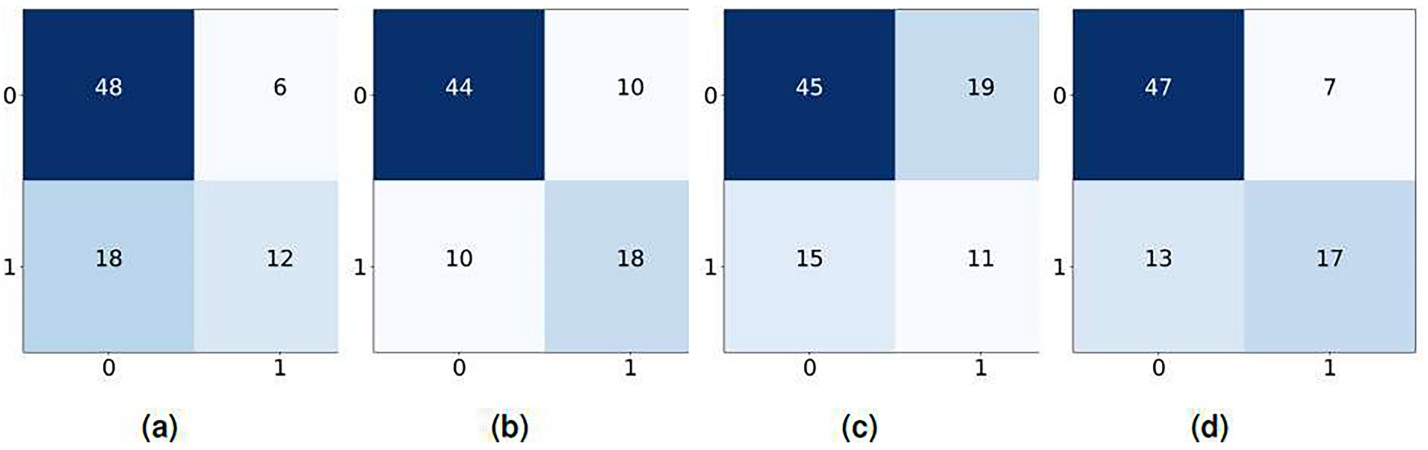

**Figure 7** Confusion matrix using glove features, (A) KNN, (B) LR, (C) NB, and (D) RF.

The confusion matrices in Fig. 7 illustrate the performance of the machine learning model on a binary classification task using GloVe features. KNN correctly predicted 60 instances and misclassified 24, while LR performed slightly better, with 62 correct predictions and 20 misclassifications. NB showed relatively lower performance, with 56 correct predictions and 34 incorrect predictions. RF performed the best, achieving 64 correct predictions and 20 misclassifications, tied in misclassification count with LR but surpassing it in incorrect classifications.

### Deep learning model performance for allergen detection

Table 15 compares the performance of four deep learning models (LSTM, GRU + RNN, CNN + LSTM, and GRU + CNN + RNN) on allergen food detection, based on accuracy, precision, recall, and F1-score. The GRU + RNN model achieves the highest accuracy (0.98), likely due to GRU's ability to capture long-term dependencies in sequential data, and RNN's effectiveness in handling temporal relationships. LSTM follows with a high

**Table 15  Allergen detection results using deep learning models.**

| Classifier | Wall time | Accuracy | Class | Precision | Recall | F1-score |
|---|---|---|---|---|---|---|
| LSTM | 30 s | 0.96 | Contains | 1.00 | 0.94 | 0.97 |
| | | | Does not Contains | 0.92 | 1.00 | 0.96 |
| | | | Micro Avg. | 0.96 | 0.97 | 0.96 |
| | | | Weighted Avg | 0.97 | 0.96 | 0.96 |
| GRU + RNN | 36.2 s | 0.98 | Contains | 1.00 | 0.96 | 0.98 |
| | | | Does not Contains | 0.94 | 1.00 | 0.97 |
| | | | Micro Avg. | 0.97 | 0.98 | 0.97 |
| | | | Weighted Avg | 0.98 | 0.98 | 0.98 |
| CNN + LSTM | 24.2 s | 0.90 | Contains | 0.92 | 0.92 | 0.92 |
| | | | Does not Contains | 0.88 | 0.88 | 0.88 |
| | | | Micro Avg. | 0.90 | 0.90 | 0.90 |
| | | | Weighted Avg | 0.90 | 0.90 | 0.90 |
| GRU + CNN + RNN | 40.9 s | 0.95 | Contains | 0.98 | 0.94 | 0.96 |
| | | | Does not Contains | 0.91 | 0.97 | 0.94 |
| | | | Micro Avg. | 0.95 | 0.95 | 0.95 |
| | | | Weighted Avg | 0.95 | 0.95 | 0.95 |

accuracy of 0.96, as it is also adept at handling sequential data but typically performs slightly worse than GRU models in this context. The CNN + LSTM model, which combines convolutional and sequential processing, has the lowest accuracy (0.90), likely because CNNs are more suited for spatial tasks, and the sequential nature of the problem isn't fully leveraged. The GRU + CNN + RNN model performs well with an accuracy of 0.95, but it doesn't surpass the simpler GRU + RNN model, possibly due to added complexity and less optimal tuning. Overall, the GRU + RNN model stands out as the most effective for allergen detection in text-based data due to its ability to maintain relevant context across sequences. Machine learning models outperform deep learning approaches, especially in binary classification tasks (*Sokoliuk et al., 2020*), due to data requirements and model complexity differences. Deep learning models typically require larger datasets to learn effectively, and their performance may degrade with limited data due to overfitting or insufficient feature learning. Conversely, machine learning models can excel with smaller datasets, particularly with well-engineered features. Additionally, factors such as imbalanced data, suboptimal hyperparameter tuning, unsuitable architectures, and insufficient training time can hinder the performance of deep learning models. In our case dataset is small and has a smaller feature set, so the performance is lower compared with machine learning algorithms. These challenges emphasize the importance of matching the model complexity to the dataset characteristics and task requirements. The results shown in Fig. 8 compare the performance of various deep learning models like LSTM, GRU + RNN, CNN + LSTM, and GRU + RNN + CNN across metrics such as accuracy, precision, recall, and F1-score. All models perform consistently well, with scores ranging from 0.9 to 0.98. GRU + RNN + CNN and CNN + LSTM show slightly higher metrics, particularly in accuracy and precision, reaching up to 0.98, indicating their superior ability to capture and

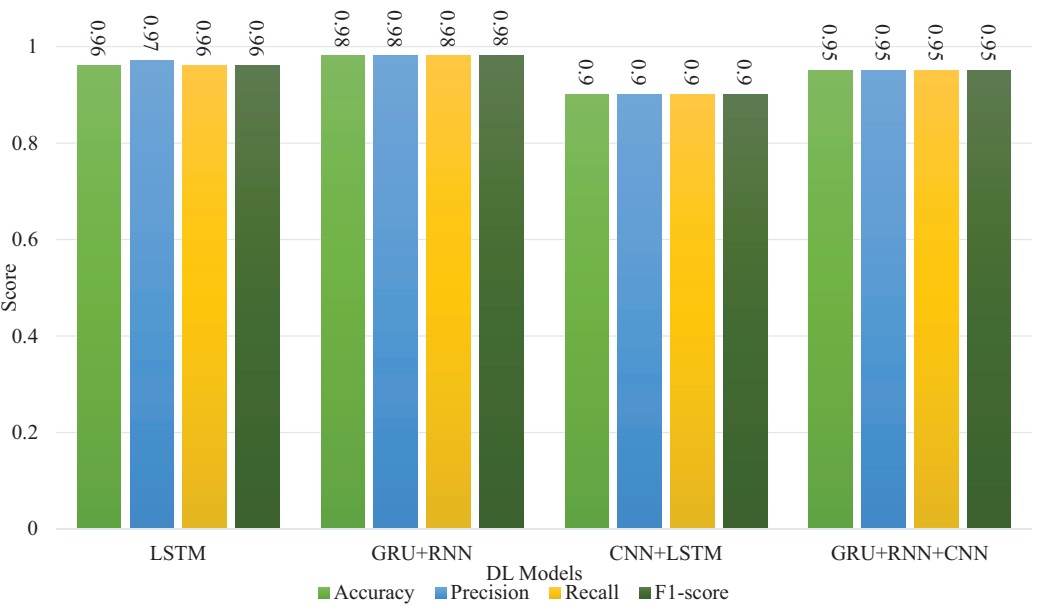

**Figure 8 Deep learning models performance comparison.**

**Table 16 K-fold cross-validation results for deep learning.**

| Approach | Mean (±Std) | Wall time |
|---|---|---|
| LSTM | 0.96 (±0.02) | 31.6 s |
| GRU + RNN | 0.98 (±0.02) | 49.7 s |
| CNN + LSTM | 0.90 (±0.05) | 37.62 s |
| GRU + CNN + RNN | 0.95 (±0.03) | 56.11 s |

process complex patterns. Overall, these results highlight the robustness of hybrid architectures in improving performance across classification tasks.

Table 16 presents the performance of various deep learning models for food allergen classification. The evaluation metric used is the mean accuracy, which reflects the average classification performance over multiple runs. The standard deviation (±std) indicates the variability or consistency of the model's performance across those runs. A lower standard deviation suggests that the model performs consistently across different runs. Among all models, the GRU + RNN architecture achieved the highest mean accuracy of 0.98 with a low standard deviation of 0.02, indicating both high accuracy and reliability. The LSTM model also performed well with a mean accuracy of 0.96. Models like CNN + LSTM and GRU + CNN + RNN showed slightly lower accuracy, possibly influenced by the fact that they were trained on a smaller subset of the dataset due to data constraints. As expected, more complex models required longer training times, as reflected in the wall time column.

The results from our 10-fold cross-validation experiments reveal valuable insights into the comparative performance of different classification models and feature representation

**Table 17 K-fold cross-validation results for machine learning.**

| Approach | TF-IDF | | BoW | | GloVe | |
|---|---|---|---|---|---|---|
| | Mean (±Std) | Wall time | Mean (±Std) | Wall time | Mean (±Std) | Wall time |
| RF | 0.91 (±0.09) | 1.62 ms | 0.91 (±0.09) | 1.52 ms | 0.73 (±0.09) | 1.75 ms |
| KNN | 0.81 (±0.08) | 51.7 ms | 0.83 (±0.13) | 83.6 ms | 0.69 (±0.06) | 86.4 ms |
| LR | 0.86 (±0.07) | 142 ms | 0.91 (±0.11) | 132 ms | 0.68 (±0.12) | 149 ms |
| NB | 0.86 (±0.07) | 128 ms | 0.91 (±0.11) | 149 ms | 0.68 (±0.12) | 138 ms |

techniques as shown in Table 17. LR combined with BoW emerged as the most effective approach, achieving the highest accuracy of 0.91 (±0.11). This performance is comparable to that of the more complex RF model, which scored 0.91 (±0.09) using BoW. However, LR has the added advantages of being a simpler, more interpretable model with a probabilistic foundation, making it well-suited for many practical applications. Notably, LR outperformed its own variants when paired with TF-IDF (0.86 ± 0.07) and GloVe embeddings (0.68 ± 0.12), indicating that BoW provided the most effective representation for the dataset in question. Similarly, NB achieved the same accuracy as LR with BoW (0.91 ± 0.11), but LR is generally more robust and flexible, especially for imbalanced or complex data distributions. KNN, while showing reasonable performance with BoW (0.83 ± 0.13), lagged behind LR in both accuracy and computational efficiency.

Across all models, GloVe embeddings resulted in lower performance, suggesting that pre-trained word embeddings may not align well with the domain-specific vocabulary or context of the dataset. This further supports the effectiveness of simpler, *corpus*-driven methods like BoW in certain classification tasks. Regarding computational efficiency, wall time comparisons show that LR with BoW offers a practical balance. Although it is marginally slower than RF (132 *vs.* 1.52 ms per fold), the trade-off is justified given LR's interpretability and comparable accuracy. Furthermore, LR is significantly more efficient than KNN, which requires 83.6 to 86.4 ms per fold yet delivers lower accuracy. These results confirm that LR with BoW is not only highly accurate but also computationally feasible and interpretable, making it a strong choice for text classification tasks in real-world settings.

## Performance evaluation of the proposed model through real-time testing

In the online testing phase, we employed OCR using the EasyOCR package to extract text from an image. A total of 20 images per class were captured in real time and were processed to extract multiple blocks of text, including phrases such as "Low FAT VANILLA FLAVOURED YOGHURT" and "INGREDIENTS: Skim Milk, Concentrated." Images of food product labels were captured using a Samsung Galaxy S21 smartphone equipped with a 12 MP rear camera, under typical indoor lighting conditions (300–500 lux). The device was held at a fixed distance of approximately 20 cm from the product surface to minimize distortion. To ensure consistency, images were taken against a plain background with minimal ambient glare. The EasyOCR library (version 1.7.2) was used for

**Table 18 Performance evaluation metrics for real time testing.**

| Classifier | Wall time | Accuracy | Class | Precision | Recall | F1-score |
|---|---|---|---|---|---|---|
| Real time | 25 ms | 0.91 | Contains | 0.91 | 0.91 | 0.91 |
| | | | Does not Contains | 0.90 | 0.89 | 0.91 |
| | | | Micro Avg. | 0.91 | 0.91 | 0.91 |
| | | | Weighted Avg | 0.91 | 0.91 | 0.91 |

text extraction, running in a Python 3.8 environment. To assess robustness, label images were captured under three lighting conditions: natural daylight, fluorescent indoor lighting, and low light (below 150 lux) environments. Lighting-induced inconsistencies were addressed by converting images to grayscale and applying histogram equalization using OpenCV to normalize contrast. This preprocessing improved OCR text detection accuracy by reducing the impact of uneven illumination and shadow artifacts.

We evaluated the OCR system using product labels with font sizes ranging from 6 to 18 pt. EasyOCR maintained an accuracy of character recognition of 90% for fonts 10, with a minor decline (5%) in recognition for fonts below 8 pt. Labels with larger fonts showed near perfect OCR output. These results confirm the model's robustness across practical font variations commonly found on commercial packaging. Once the text was extracted, it was concatenated into a single string and split into rows using commas as delimiters. These rows were further cleaned by removing extra whitespace and were then compiled into a new DataFrame. For preprocessing, the following steps were performed: the text was converted to lowercase, HTML tags and URLs were removed, numeric values were stripped, and stopwords were eliminated. Additionally, tokenization, stemming, and lemmatization were applied to process the words. After preprocessing, an LR machine learning model was employed to predict labels based on the cleaned data. For curved product surfaces (*e.g.*, cylindrical bottles), the OCR pipeline includes a text localization step using bounding boxes to isolate text regions. These regions are corrected using affine transformations to reduce perspective distortion. The grid generator described in Fig. 2 assists in normalizing slanted or wrapped text blocks into flat 2D planes. This pre-correction significantly enhanced the readability and accuracy of OCR outputs on non-planar labels. Of the 40 images, 36 were correctly classified under the labels "Contains," and "Does not Contains" indicating the presence of allergenic ingredients in the respective products with approx. 91% accuracy scores. In contrast, only one image was incorrectly predicted. The classification report and confusion matrix are presented in Table 18 and Fig. 9, respectively. This result further validates the effectiveness of the proposed model as a reliable tool for allergen prediction in various products.

The performance of the four deep learning models varies across epochs as shown in Fig. 10. All models achieve rapid convergence in training mean square error (MSE), with near-zero values indicating effective learning. However, validation MSE fluctuates significantly for LSTM and GRU-RNN, suggesting instability in generalization, while CNN-LSTM and GRU-CNN-RNN show consistently higher validation MSE, highlighting generalization challenges despite minimal fluctuations. These trends indicate potential

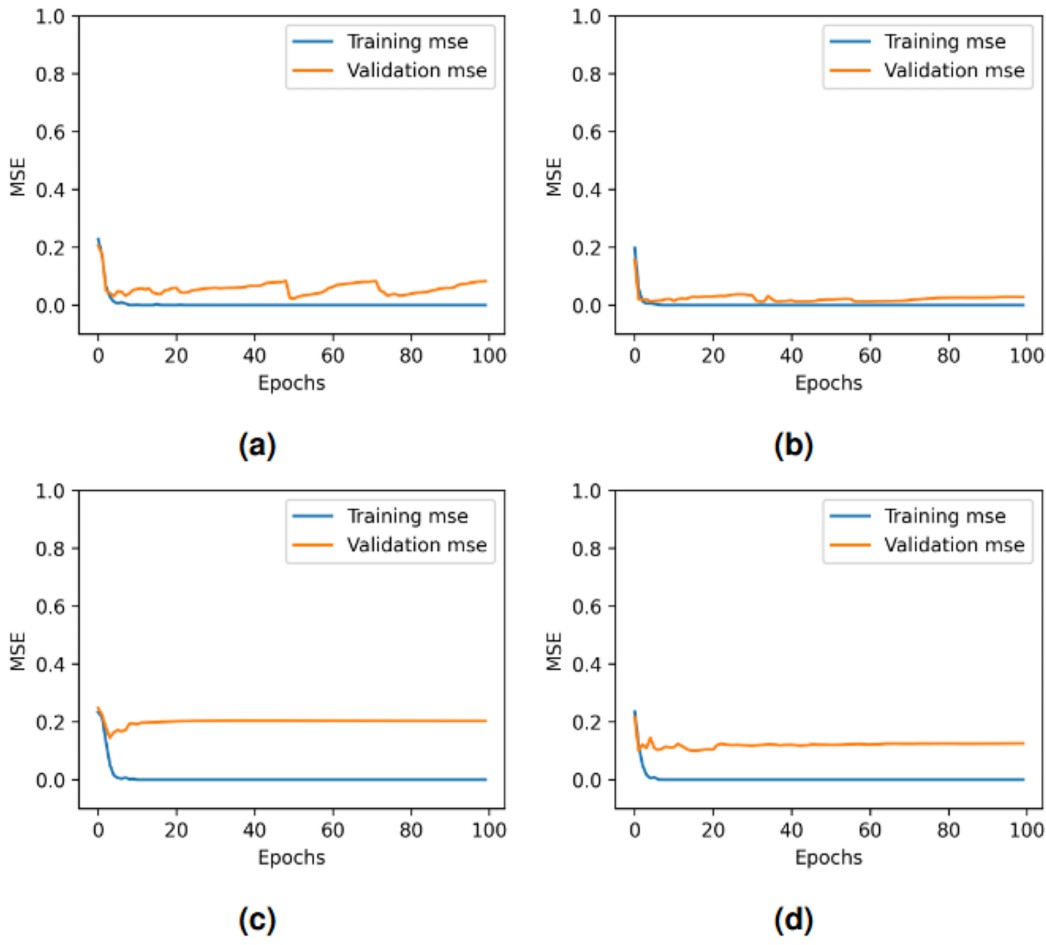

**Figure 9** Per epoch performance graphs for deep learning models: (A) LSTM, (B) GRU + RNN, (C) CNN + LSTM, and (D) GRU + CNN + RNN.   

overfitting or complexity issues affecting validation performance across models. In our proposed framework, through real-time testing, as shown in Table 19, the system successfully detected allergen-related terms (*e.g.*, "nuts") while avoiding false positives on allergen-free items. This visual demonstration supports the quantitative findings reported earlier and reflects the practical utility of the proposed approach in consumer-facing applications.

## Comparison with other state-of-the-art models

To show the importance of our recommended plan more clearly, we have compared its results with some recent top-notch systems. Table 20 shows how well the new model beats the known best methods in Allergy classification found so far. It is clear from this comparison that our proposed idea works better than current top-notch literature techniques for classifying allergens. It also shows the given model can find allergens with 99% accuracy on average. This is better than other advanced techniques in studies.

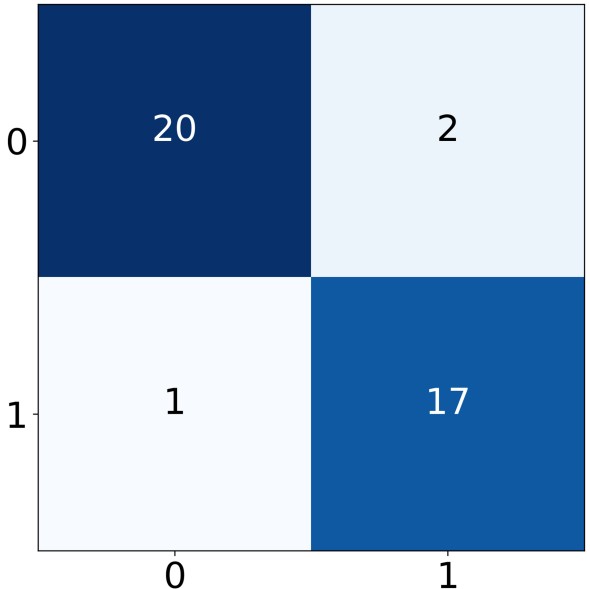

**Figure 10 Confusion matrix for real-time model testing.**

---

**Table 19  OCR results and allergen classification in real-time food label analysis.**

| Image | Extracted OCR text | Predicted label | Ground truth |
|---|---|---|---|
| | **PARBOILED RICE** | No allergens detected | Correct (No allergen) |
| | Energy: 117 kcal | | |
| | Salt: <0.01g | | |
| | Fat: <0.5 g | | |
| | **LIDL product** | Likely contains allergens | Correct (Possible allergen) |
| | Fat: 48.8 g | | |
| | Saturates: 8.5 g | | |
| | Protein: 21.3 g | | |
| | **Omega-3: 9.9 g** | Contains allergens (nuts) | Correct (Nut allergen) |
| | Protein: 15.5 g | | |
| | Note: may contain nuts | | |

**Table 20 Comparative analysis with existing studies.**

| Ref | Model | Performance |
|---|---|---|
| *Ktona et al. (2022)* | Part | 85.99% |
| *Kumar & Rana (2023)* | ETC + DBN + CatBoost | 89.16% |
| *Omurca et al. (2019)* | (DT + SVM + LR + kNN) | 77% |
| *Nedyalkova et al. (2023)* | SVM | 93% |
| *Wang et al. (2021)* | BERT | 93.10% |
| *Sharma et al. (2021)* | RF | 83.39% |
| *Roither, Kurz & Sonnleitner (2022)* | LR with TF-IDF | 93% |
| *Mishra et al. (2022)* | YOLOR with interference speed of 8.4 ms | 78.29% |
| *Rohini et al. (2021)* | VGG16 | 97.37% |
| Proposed | LR with wall time 13 ms | 99% |

**Table 21 Performance evaluation of deep learning models using batch size 128 and varying learning rates.**

| Classifier | Batch size | Wall time | Accuracy | Class | Precision | Recall | F1-score |
|---|---|---|---|---|---|---|---|
| LSTM | 128 | 12.8 s | 0.88 | Contains | 0.91 | 0.91 | 0.91 |
| | | | | Does not Contains | 0.88 | 0.88 | 0.88 |
| | | | | Micro Avg. | 0.90 | 0.90 | 0.90 |
| | | | | Weighted Avg | 0.90 | 0.90 | 0.90 |
| GRU + RNN | 128 | 36.2 s | 0.96 | Contains | 0.96 | 0.96 | 0.98 |
| | | | | Does not Contains | 0.94 | 1.00 | 0.97 |
| | | | | Micro Avg. | 0.96 | 0.96 | 0.96 |
| | | | | Weighted Avg | 0.96 | 0.96 | 0.96 |
| CNN + LSTM | 128 | 24.2 s | 0.91 | Contains | 0.93 | 0.93 | 0.93 |
| | | | | Does not Contains | 0.90 | 0.90 | 0.90 |
| | | | | Micro Avg. | 0.91 | 0.91 | 0.91 |
| | | | | Weighted Avg | 0.91 | 0.91 | 0.91 |
| GRU + CNN + RNN | 128 | 40.9 s | 0.94 | Contains | 0.94 | 0.94 | 0.94 |
| | | | | Does not Contains | 0.92 | 0.92 | 0.92 |
| | | | | Micro Avg. | 0.94 | 0.94 | 0.94 |
| | | | | Weighted Avg | 0.94 | 0.94 | 0.94 |

While previous allergen datasets have contributed valuable insights, they are often limited in scope, containing only basic ingredient and allergen labels without accounting for complex food compositions such as sweeteners, fats, oils, and seasonings. These missing components may play a critical role in allergenic responses but are often overlooked, reducing the dataset's effectiveness for comprehensive analysis or real-time application. It is important to note that the datasets used in the comparative studies vary significantly in terms of sample size, feature granularity, and target allergen classes. For example, the study by *Ktona et al. (2022)* used only 155 medical records focused on detecting allergies in Albania, while *Wang et al. (2021)* used large; multi-source datasets (*e.g.,* SDAP, NCBI). In contrast, our proposed model was evaluated on a balanced and

moderately sized dataset containing detailed ingredient-level attributes. Therefore, although the proposed method outperforms others in accuracy, it is essential to interpret these results within the context of dataset complexity, generalizability, and domain specificity. Our proposed system addresses these gaps by incorporating a more detailed feature set extracted *via* OCR from real product labels. This includes secondary and hidden ingredients, enabling a more nuanced and realistic allergen detection model. Furthermore, by integrating this approach into a real-time OCR-enhanced machine learning pipeline, our system not only improves detection accuracy but also ensures practical usability for consumers, distinguishing it from previous predominantly offline or small-scale efforts.

To further investigate the impact of learning rate and batch size on the performance and convergence behavior of deep learning models, we conducted additional experiments by fixing the batch size to 128 and varying the learning rate across 0.1, 0.01, and 0.02. These configurations were chosen based on standard tuning practices and to address fluctuations observed in earlier training loss curves. The performance metrics for each model under these settings including accuracy, precision, recall, and F1-score are summarized in Table 21. The results indicate that all models maintained strong classification performance, suggesting that the selected hyperparameter ranges support effective learning and reduce the likelihood of overfitting or underfitting.

# CONCLUSIONS

This study performs experiments for allergen prediction using machine learning, testing both in real-time and offline modes. The study extracts features using OCR from product images and then applies feature extraction techniques to process the data, followed by prediction models to identify allergens. Through extensive research, we conclude that state of the art (SOTA) models perform well when tested on the same dataset, but their performance drops when applied to real-time scenarios. For example, the proposed approach achieved 99% accuracy using LR on the dataset, but when we passed a product image through the OCR pipeline, the accuracy dropped to 91%. This study also found that the small size of benchmark allergen datasets makes it difficult to train deep-learning models effectively. As a result, simpler models like LR perform better with basic features, such as BoW, because the problem involves binary classification with fewer data points. LR is well-suited for classifying allergens and non-allergens using a sigmoid function.

## Limitations and future directions

While this study makes a significant contribution to real-time allergen detection, several limitations remain that provide opportunities for further improvement. One key challenge lies in enhancing the accuracy of real-time predictions, which can be impacted by OCR errors due to poor image quality, complex label designs, or spelling mistakes in extracted text. For future work, we plan to extract additional features from images to enrich the input data and further improve classification performance. We also intend to incorporate a large language model to correct ingredient spelling errors, which may arise from OCR inaccuracies and negatively affect the prediction outcomes. Specifically, we plan to expand the dataset by integrating food items from diverse and authoritative sources, such as

government allergen databases and international food regulatory bodies. This enhancement aims to improve the dataset's representativeness and the model's generalizability across regional and global food products, ultimately contributing to the novelty and broader applicability of our proposed method. Additionally, we are exploring the deployment of the proposed framework as an Android application, with the goal of enabling real-time allergen detection and improving accessibility for end users in practical settings.

## ACKNOWLEDGEMENTS

In the preparation of this manuscript, we used ChatGPT (developed by OpenAI) solely for language refinement, including grammar correction, clarity enhancement, and improving sentence structure. All core research components including the formulation of ideas, experimental design, data analysis, model development, and conclusions were independently conceived and executed by the authors without the use of AI tools. No generative AI was used to produce original scientific content, generate figures, or perform any part of the data analysis.

### Funding
The authors received no funding for this work

### Competing Interests
The authors declare that they have no competing interests.

### Author Contributions
- Erol Kına conceived and designed the experiments, performed the experiments, analyzed the data, performed the computation work, prepared figures and/or tables, authored or reviewed drafts of the article, and approved the final draft.

### Data Availability
The data and code are available in the Supplemental Files.

The code is available at GitHub and Zenodo:

- https://github.com/Arehmans/food-allergen.git.

- Arehmans. (2025). Arehmans/food-allergen: Initial Release for Publication (v1.0). Zenodo. https://doi.org/10.5281/zenodo.16404339.

### Supplemental Information
Supplemental information for this article can be found online at http://dx.doi.org/10.7717/peerj-cs.3338#supplemental-information.

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
