# Peer review of "Real-time food allergen detection using OCR-enhanced machine learning techniques"

_PeerJ Computer Science, doi:10.7717/peerj-cs.3338_

## Round 0.1 · original submission · Minor Revisions

· Academic Editor

Minor Revisions

**Language Note:** When you prepare your next revision, please either (i) have a colleague who is proficient in English and familiar with the subject matter review your manuscript, or (ii) contact a professional editing service to review your manuscript. PeerJ can provide language editing services - you can contact us at [email protected] for pricing (be sure to provide your manuscript number and title). – PeerJ Staff

Reviewer 1 ·

Basic reporting

All core research components, including the formulation of ideas, experimental design, data analysis, model development, and conclusions, were independently conceived and executed.

Experimental design

Good insights on real-time.

Validity of the findings

A mathematical model and algorithm have been implemented on a state-of-the-art topic.

Conclusions are well stated & limited to supporting results.

Cite this review as

Reviewer 2 ·

Basic reporting

The article proposes a system designed to read ingredient lists from scanned product labels to detect the presence of allergenic components. The system processes image-based labels as input, identifying the presence or absence of allergens using machine learning (ML) and deep learning (DL) algorithms.

The problem is well framed, and the motivation for the study is clearly articulated. The central objective to explore ML and DL algorithms for reading food labels and classifying allergen content is relevant and timely.

The related work section is comprehensive, referencing several pertinent studies. Each work is summarized, and a comparative analysis is provided, detailing the algorithms used, datasets employed, and performance outcomes. The reference list is also thorough and includes recent contributions to the field.

However, the manuscript requires the following corrections and improvements:

In line 187, Table 1 must be explicitly referenced to guide the reader appropriately.

The acronym “RF” (Random Forest) should be defined upon first use to ensure clarity for all readers, particularly those less familiar with the terminology.

Experimental design

The proposed methodology is well described and applied coherently. Following data pre-processing, feature engineering is performed using Bag of Words, Global Vectors for Word Representation (GloVe), and Term Frequency–Inverse Document Frequency (TF-IDF) to extract robust features essential for training binary machine learning classifiers: Linear Regression, Random Forest, Naïve Bayes, and K-Nearest Neighbors. Deep learning models, CNN, GRU, and LSTM are also evaluated, contributing to a comprehensive methodological framework.

However, the real-time experiment is described only superficially, which significantly limits the practical understanding and reproducibility of the system. The following critical details must be added:

Procedure for capturing the label image: Clearly describe the hardware and software setup used for image acquisition.
Robustness to lighting variations: Discuss how the system performs under different lighting conditions and whether any normalization techniques are applied.

Analyze how varying font sizes affect the accuracy of text recognition.

Explain the approach used to correct or compensate for distortions caused by curved surfaces.

Additionally, it is strongly recommended to include a table with example images used in the real-time experiment. This will enhance the clarity and transparency of the experimental setup and results.

Validity of the findings

The author presents a solid discussion of the results, concluding that the LR model performs best when using features extracted with the Bag of Words technique. This conclusion is well supported by the experimental data.

However, several critical issues remain that must be addressed to meet the publication standards of the journal:

While the author compares the proposed method with other state-of-the-art models, the analysis is limited to performance metrics. A more meaningful comparison should also consider the differences in datasets used across the referenced studies. This context is essential for a fair and comprehensive evaluation of the proposed method’s effectiveness.

The contents of the two columns in Table 7 are identical. The example intended to illustrate the removal of stopwords is not evident. This table must be corrected to clearly demonstrate the effect of stopword removal.

Several figures and tables are misplaced. For instance, Figure 4 appears on page 14 but is first referenced on page 6. This disrupts the flow of the manuscript and must be corrected. A thorough review of the entire document layout is necessary to ensure that all figures and tables are appropriately positioned near their first mention in the text.

The terms “Micro Avg.” and “Weighted Avg.” appear in the results table but are not defined or explained. Without proper context, these metrics do not add value and may confuse readers. The author should either define these terms clearly or remove them if they are not essential to the analysis.

Cite this review as

·

Basic reporting

The manuscript covers all the required information, including the introduction, which is subject to the research question and intent. In fact, the manuscript also has a detailed level of limitations and gaps in the studies within the food industry. I liked the proposed methodology mentioned in this manuscript.

Experimental design

The methodology is well-structured and aligns with current best practices, ensuring the research is both robust and reliable. Additionally, the author's insights into potential future research avenues are particularly thought-provoking and could significantly contribute to the field.

This manuscript covers the experimental design as well as the architectural diagram of the proposed system to manage the problem statement. The Optical Character Recognition (OCR) was employed to extract textual information from the images, facilitating accurate allergen identification as shown in Fig. 2 and Tables 1, 2.

Validity of the findings

It discusses the validity of the main conclusions and provides output graphs and tables that clearly show the study's results. These visual aids simplify the data, allowing readers to grasp complex information at a glance. Additionally, the analysis highlights potential areas for further research, emphasizing the importance of ongoing inquiry in the field.

Additional comments

Please kindly upload good quality tables 1,2,3,4,5, and figures 2, 4.

---

## Round 0.2 · accepted · Accept

· Academic Editor

Accept

Dear Authors,

One of the reviewers who requested minor revisions in the previous revision did not respond to the revision within the requested time frame. The other reviewer accepts your article as it stands. Your paper is sufficiently improved and ready for publication.

Best wishes,